# Health insurance non-enrollment among women in Sierra Leone: A cross-sectional analysis of the 2019 Demographic and Health Survey

Augustus Osborne[1]*, Mainprice Akuoko Essuman[2,3], Peter Bai James[4,5], Camilla Bangura[1], Richard Gyan Aboagye[6], Comfort Z. Olorunsaiye[7], Abdul-Aziz Seidu[8], Bright Opoku Ahinkorah[9,10]

**1** Department of Biological Sciences, School of Environmental Sciences, Njala University, PMB, Freetown, Sierra Leone, **2** Department of Medical Laboratory Science, School of Allied Health Sciences, College of Health and Allied Sciences, University of Cape Coast, Central, Ghana, **3** Department of Biological Sciences, Southern Illinois University Edwardsville, Edwardsville, Illinois, United States of America, **4** National Centre for Naturopathic Medicine, Faculty of Health, Southern Cross University, Lismore, Australia, **5** Faculty of Pharmaceutical Sciences, College of Medicine and Allied Health Sciences, University of Sierra Leone, Freetown, Sierra Leone, **6** Department of Family and Community Health, Fred N. Binka School of Public Health, University of Health, and Allied Sciences, Hohoe, Ghana, **7** Department of Public Health, Arcadia University, Glenside, Pennsylvania, United States of America, **8** Public Health and Tropical Medicine, James Cook University, Townsville, Australia, **9** REMS Consultancy Services, Takoradi, Sekondi-Takoradi, Ghana, **10** Faculty of Health and Medical Sciences, The University of Adelaide, Adelaide, Australia.

* augustusosborne2@gmail.com

## Abstract

### Background

Health insurance enrollment is a vital component of universal health coverage and access to essential healthcare services. However, in Sierra Leone, enrollment remains persistently low, posing a major public health challenge. Women of reproductive age (15–49 years) represent a critical population for health insurance enrollment due to their unique healthcare needs, particularly related to reproductive health, pregnancy, and childcare. Despite their importance, women face notable barriers to health insurance enrollment, including financial constraints, gender inequalities, and sociocultural challenges. This study examines the factors associated with health insurance non-enrollment among women aged 15–49 in Sierra Leone.

### Methods

We analysed data from a weighted sample of 15,574 women aged 15–49 years from the 2019 Sierra Leone Demographic and Health Survey. Percentages were used to present the proportion of health insurance non-enrollment among the women. Multivariable binary logistic regression analysis was used to examine the factors associated with health insurance non-enrollment among the women.

**Data availability statement:** The data set is openly available upon permission from the MEASURE DHS website https://www.dhsprogram.com/data/available-datasets.cfm

**Funding:** The author(s) received no specific funding for this work.

**Competing interests:** The authors have declared that no competing interests exist.

**Abbreviations:** EA, enumeration area; AOR, adjusted odds ratio; CI, confidence interval; COR, crude odds ratio; DHS, demographic and health survey; SLDHS, sierra leone demographic health survey; OR, odds ratio; SDG, sustainable development goals; VIF, variance inflation factor.

## Results

The proportion of health insurance non-enrollment among the women was 96.02%. Women with higher education were less likely to be uninsured (adjusted odds ratio [aOR]: 0.35, 95% CI: 0.18–0.64) compared to those with no education. Being employed also reduced the odds of being uninsured (aOR: 0.47, 95% CI: 0.36–0.62) compared to women who were employed. Listening to the radio less than once a week was associated with lower odds of being uninsured (aOR: 0.71, 95% CI: 0.53–0.97) compared to women who did not listen to the radio at all. On the other hand, women who reported distance to a health facility as a big problem were more likely to be uninsured (aOR: 2.21, 95% CI: 1.03–4.75) compared to those who did not consider it a problem. Regionally, women living in the Northwestern (aOR: 0.07, 95% CI: 0.03–0.14) and Northern (aOR: 0.28, 95% CI: 0.12–0.66) regions were less likely to be uninsured compared to those residing in the Eastern region.

## Conclusions

Health insurance non-enrollment was high among women in Sierra Leone. Education, employment, and exposure to listening to the radio were associated with increased health insurance enrollment, highlighting the need to address socioeconomic barriers and leveraging mass media campaigns to educate women on the importance of getting covered by health insurance. Geographic and regional disparities in health insurance enrollment underscore the importance of improving healthcare accessibility and implementing targeted, community-based interventions to promote health insurance uptake. Also, subsiding health insurance subscription fee could increase its enrollment, especially among women from low socioeconomic households.

## Introduction

In recent years, global health policy has increasingly focused on achieving universal health coverage (UHC) as a means of ensuring equitable access to healthcare services and protecting individuals from out-of-pocket (OOP) healthcare expenses [1–3]. Health financing remains a significant challenge worldwide, particularly in low- and middle-income countries (LMICs), where high OOP expenditures often hinder access to essential healthcare services and push millions of households into poverty each year [4]. To address these challenges, health insurance has emerged as a critical mechanism to improve access to healthcare, reduce financial barriers, and promote equity in health systems [1]. Despite its potential, health insurance coverage remains uneven globally, with significant gaps in enrollment, particularly among vulnerable populations such as women, children, and those in low-income settings [5].

In sub-Saharan Africa (SSA), health insurance policies have been implemented over the past two decades as part of broader efforts to achieve UHC and meet the Sustainable Development Goals (SDGs), particularly SDG 3.8 (achieving UHC) and SDG 3.1 (reducing maternal mortality) [6]. Countries such as Ghana, Rwanda,

Kenya, Nigeria, and South Africa have introduced various health insurance programmes to improve healthcare accessibility and reduce financial strain on their populations [7]. These programmes have targeted both formal sector employees and informal sector workers, with some extending coverage to entire communities. Despite these efforts, enrollment in several other countries in SSA remains low, with significant disparities in coverage between urban and rural populations, and between men and women [8]. Women, particularly those of reproductive years, face barriers to enrollment, including economic constraints, limited awareness, and sociocultural factors that restrict their access to healthcare services [9].

Sierra Leone, a low-income country in SSA with a population of approximately 7.9 million, exemplifies these challenges. Women, who constitute 51% of the population, bear a disproportionate burden of healthcare needs, particularly related to maternal and reproductive health [10]. The country continues to face a high maternal mortality ratio, recorded at 717 deaths per 100,000 live births in 2019, and a high adolescent birth rate of 102 births per 1,000 women aged 15–19 [11]. Health insurance is a vital strategy for ensuring universal access to affordable and quality healthcare, particularly for vulnerable groups like women and children [12]. While maternal health services are often prioritized in low-income countries such as Sierra Leone due to high maternal mortality rates, health insurance schemes typically aim to provide broader coverage, including universal and primary health care services [12]. These services encompass preventative care, treatment for various diseases, and essential health interventions. In Sierra Leone, health insurance supports the broader goal of universal health coverage, making it crucial to address barriers to enrollment among women to improve overall health outcomes and advance equity in healthcare access [12]. However, Sierra Leone's health system is weak and underfunded, with inadequate infrastructure, human resources, and supplies [13]. The country also lacks a comprehensive and sustainable health financing mechanism to protect people from catastrophic health expenditures and ensure equitable access to health care [13].

According to the WHO, only 5.6% of women and 3.9% of men above the statutory pensionable age receive a pension in Sierra Leone [13]. Moreover, only 53% of women of reproductive age have their need for family planning satisfied with modern methods [11]. To mitigate the issue of financial inaccessibility, the Free Healthcare Initiative was implemented in 2010 to eliminate user fees for pregnant and breastfeeding mothers and children under five [14]. In addition, the Ministry of Health and Sanitation has initiated the execution of the recently formulated National Health Sector Strategic Plan from 2017 to 2021. In 2018, the government introduced the Sierra Leone Social Health Insurance (SLeSHI) programme to enhance the financial accessibility of healthcare services [15]. However, these services have not been fully functional as they rely heavily on donor funding and lack adequate infrastructure, thus having low enrolment rates among the population.

Previous research on health insurance in Sierra Leone has focused on various aspects, such as the willingness to pay for health insurance in the informal sector [15] and population characteristics influencing the benefits basket of the national social health insurance scheme [16]. Health insurance has been shown to improve health outcomes and reduce poverty among women and their families in Sierra Leone [12]. It reduces out-of-pocket payments for healthcare, increases the utilisation of preventive and curative services, and enhances the quality and efficiency of healthcare delivery [17]. However, health insurance coverage in Sierra Leone remains alarmingly low, with only 4% of women and 3% of men enrolled, according to the 2019 Sierra Leone Demographic and Health Survey (SLDHS) [11]. This translates to just 3.5% of the population having access to health insurance, highlighting the critical lack of financial protection for healthcare [11]. While women have a slightly higher enrollment rate than men, this marginal difference does not negate the significant barriers to enrollment faced by both genders. Women in Sierra Leone encounter unique social, economic, and cultural challenges, such as lower income levels, reduced decision-making power, and limited access to information about health insurance schemes [12]. These factors exacerbate disparities in healthcare access and financial security, making the low overall coverage and the gendered barriers to enrollment a pressing issue for policymakers and researchers alike.

Despite the evidence on the potential benefits of health insurance, there is limited research examining the factors contributing to the low enrollment of women in health insurance schemes in Sierra Leone. Existing studies have largely

focused on willingness to pay or general population characteristics without delving into the gender-specific barriers that women face [15,16]. Furthermore, while health insurance is widely recognized as a pro-poor measure, there is a lack of comprehensive analysis on its gendered impact and the specific challenges women encounter in accessing health insurance in low-income settings like Sierra Leone. This study addresses this gap by exploring the factors influencing the low enrollment of women in health insurance using the 2019 SLDHS. The findings will contribute to the existing literature by providing gender-specific insights and evidence-based recommendations to improve health insurance coverage and, ultimately, the health and well-being of women and their families in Sierra Leone.

## Methods

### Data source and design

The 2019 SLDHS used for this study is part of a series of periodic cross-sectional surveys designed to collect data on demographic, health, and nutritional issues among men, women, and children. The most recent 2019 SLDHS was conducted over four months, from May 2019 to August 2019 [18]. The survey employed a stratified, two-stage cluster sampling design. In the first stage, 578 enumeration areas (EAs) were selected, comprising 214 urban and 364 rural regions. In the second stage, 24 households were systematically selected from each EA, resulting in a total sample size of 13,872 households [18]. For this study, the individual recode (IR) dataset from the 2019 SLDHS was used. This dataset is specifically designed to provide detailed information on women aged 15–49, including their sociodemographic characteristics, health indicators, and access to healthcare services. The IR dataset was selected because it is the most appropriate for analyzing factors influencing women's health insurance enrollment. A total of 15,574 women aged 15–49 were included in our study. Detailed information regarding the sampling technique and survey methodology can be found in the final DHS report [18]. We got formal authorisation to use the 2019 SLDHS data by following the prescribed procedures outlined on the official DHS programme website [19]. This study was conducted in accordance with the Strengthening Reporting of Observational Studies in Epidemiology (STROBE) guidelines [20].

### Variables

**Outcome variable.** The outcome variable in this study was health insurance coverage, and it was derived from the question, "Are you covered by any health insurance?" with the original response options coded as 1 = "Yes" and 0 = "No." For the purpose of our analysis, we recoded the variable to reflect health insurance non-enrollment, where 1 = "No" (not enrolled) and 0 = "Yes" (enrolled).

**Exposure variables.** We included sixteen exposure variables in the study, selected based on their availability in the DHS dataset and their association with health insurance coverage in prior research [5,21–23]. These variables were the age of the women, which was categorised into seven 5-year groups: 15–19, 20–24, 25–29, 30–34, 35–39, 40–44, and 45–49. Each age group was coded numerically from 1 to 7, respectively. Education level was categorized based on the highest level of formal education attained by the respondent: no education (1), primary (2), secondary (3), and higher education (4). Employment status was determined based on respondents' self-reported engagement in income-generating activities during the survey period, categorized as unemployed (0) and employed (1).

The wealth index, a composite measure of a household's cumulative living standard calculated by the DHS using principal component analysis, was divided into quintiles: poorest (1), poorer (2), middle (3), richer (4), and richest (5). This index considers household ownership of selected assets, housing construction materials, and access to water and sanitation facilities. Sex of household head was categorized as male (1) or female (2), while marital status was classified into six categories: never in union (1), married (2), cohabiting (3), widowed (4), divorced (5), and separated (6), based on respondents' self-reported marital status.

The total number of children ever born was categorized into three groups: zero births (0), one to three births (2), and four or more births (3). We also assessed problems with getting medical help for oneself across three dimensions: getting

permission to go, obtaining the money needed for treatment, and distance to the health facility. Each dimension was categorized as a big problem (1) or not a big problem (2), based on respondents' responses.

Frequency of media exposure was measured by how often respondents read newspapers or magazines, listened to the radio, or watched television. This was categorized as not at all (1), less than once a week (2), and at least once a week (3). The place of residence was classified as urban (1) or rural (2), based on the DHS classification of the respondent's living area. Lastly, region was categorized into five geographic areas of Sierra Leone: Eastern (1), Northwestern (2), Northern (3), Southern (4), and Western (5).

### Data analyses

We used descriptive and inferential statistics to examine the prevalence and factors associated with health insurance non-enrollment. First, we calculated the distribution of health insurance coverage across the explanatory variables using percentages. After this, we checked for multicollinearity among the explanatory variables using the variance inflation factor (VIF), and the results showed no evidence of high collinearity (Mean VIF = 1.98, maximum VIF = 4.64, and minimum VIF = 1.05). Next, we performed a bivariate logistic regression analysis to examine the independent associations between the explanatory variables and health insurance non-enrollment. The results were presented using crude odds ratios (cOR) with their respective 95% confidence intervals (CI). Finally, we fitted a multivariable logistic regression analysis to examine the factors associated with health insurance non-enrollment coverage by including all the explanatory variables that were significant at the crude level. The multivariable logistic regression analysis results were presented as adjusted odds ratios (aOR) with their corresponding 95% CIs. To ensure the robustness of our model, we assessed its goodness-of-fit using the Hosmer-Lemeshow test, which confirmed that the model was a good fit for the data (p > 0.05). Weighted analysis was conducted using the "svyset" command in Stata version 18 to account for disproportionate sampling, non-response, and the complex survey design of the DHS data. Statistical significance was set at p-values less than 0.05 and only variables with p-values below this threshold were included in the multivariable regression analysis.

### Ethical considerations

For our study, the dataset was accessed from the MEASURE DHS website after obtaining appropriate permissions, and no additional ethical clearance was required as this was a secondary analysis of de-identified, publicly available data.

### Results

Fig 1 shows the proportion of women with no health insurance enrollment. The results showed that 96.02% of the women had no health insurance enrollment.

### Distribution of health insurance non-enrollment across explanatory variables

Table 1 shows the results of the distribution of women who had no health insurance enrollment across the background characteristics. Younger women (aged 15–19) exhibit the highest proportion of non-enrollment (97.09%). Women with no education had the highest health insurance non-enrollment (96.44%). Employment status reveals that unemployed women had the highest health insurance non-enrollment rate of 97.25%. Women in the poorest wealth index had the highest health insurance non-enrollment rate at 98.12%. In terms of residence, rural women had the highest health insurance non-enrollment rate at 96.99%. Regionally, the Eastern region had the highest non-enrollment rate at 99.09%.

### Factors associated with health insurance non-enrollment among the women in Sierra Leone

The results in Table 2 shows the results of the factors associated with health insurance non-enrollment among women in Sierra Leone. Women with higher education were less likely to be uninsured (aOR: 0.35, 95% CI: 0.18–0.64) compared to those with no education. Employed women had reduced odds of being uninsured (aOR: 0.47, 95% CI: 0.36–0.62)

## Non-enrollment of health insurance

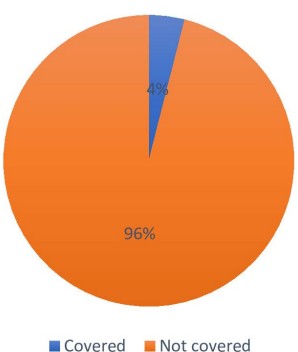

Covered ■ Not covered

**Fig 1. Proportion of women who had no health insurance enrollment in Sierra Leone.**

compared to those who were employed. Women who listened to the radio less than once a week (aOR: 0.71, 95% CI: 0.53–0.97) were less likely to be uninsured compared to those who did not listen at all. On the other hand, women who reported distance to a health facility as a big problem were more likely to be uninsured (aOR: 2.21, 95% CI: 1.03–4.75) compared to those who did not consider it a problem. Women living in the Northwestern (aOR: 0.07, 95% CI: 0.03–0.14) and Northern (aOR: 0.28, 95% CI: 0.12–0.66) regions were less likely to be uninsured compared to those in the Eastern region.

## Discussion

We examined the factors influencing non-enrollment in health insurance among women in Sierra Leone. Our results showed that only 3.98% of women in Sierra Leone are covered by health insurance. Our result is similar to the low enrollments earlier observed in Sierra Leone [22,24]. However, it is lower than the 8.5% found among countries in SSA [5]. Many women in Sierra Leone may not know how to enroll, what services are covered, and how to access them [25]. The high cost of premiums and copayments make health insurance unaffordable for many women, especially those who are poor, unemployed, or informal workers [26]. Low quality and accessibility of health services, especially in rural areas, discourage women from seeking care or using their insurance benefits [12,27].

In our study, women with higher education had lower odds of non-enrollment for health insurance coverage than women without education in Sierra Leone. Our finding is similar to the findings of Amu et al. [5] in SSA and Aboagye et al. [28] in Ghana who found that enrollment in health insurance is lower among uneducated women. Women with higher education may have more income and savings to afford the cost involved in obtaining health insurance [29]. Again, women with higher education may have more access to information and awareness of the benefits and risks of health insurance.

In our study, employed women had lower odds of non-enrollment than those unemployed in Sierra Leone. Our study is in line with the findings of Amu et al. [5] who found that women with employment had higher odds of health insurance coverage. Employed women may have more access to information and awareness about health insurance benefits and how to enroll than unemployed women [30–32]. Employed women may have more incentives and opportunities to enroll in health insurance schemes offered by their employers than unemployed women [30,31].

Our study found that women who listen to the radio less than once a week have significantly lower odds of non-enrollment in health insurance compared to those who do not listen to the radio at all. This finding suggests that listening to the radio, regardless of frequency, may increase awareness and understanding of health insurance, potentially facilitating enrollment. This aligns with findings from a previous study conducted in SSA [5], which highlights the role of media exposure in promoting health-related behaviors. Women who listen to the radio may be exposed to health messages or

**Table 1. Distribution of health insurance coverage across women's characteristics.**

| Variables | Weighted frequency (%) | Health insurance coverage | |
|---|---|---|---|
| | | Yes (%) | No (%) |
| **Age in 5-year groups** | | | |
| 15-19 | 3427 (22.00) | 2.91 | 97.09 |
| 20-24 | 2629 (16.88) | 3.77 | 96.23 |
| 25-29 | 2728 (17.51) | 4.62 | 95.38 |
| 30-34 | 1942 (12.47) | 4.90 | 95.10 |
| 35-39 | 2224 (14.28) | 4.19 | 95.81 |
| 40-44 | 1337 (8.58) | 3.84 | 96.16 |
| 45-49 | 1288 (8.27) | 4.36 | 95.64 |
| **Highest educational level** | | | |
| No education | 7081 (45.47) | 3.56 | 96.44 |
| Primary | 2103 (13.50) | 2.57 | 97.43 |
| Secondary | 5724 (36.75) | 3.83 | 96.17 |
| Higher | 666 (4.28) | 14.30 | 85.70 |
| **Current marital status** | | | |
| Never in union | 5058 (32.47) | 3.57 | 96.43 |
| Married | 9107 (58.48) | 4.26 | 95.75 |
| Cohabiting | 608 (3.90) | 2.62 | 97.38 |
| Widow | 351 (2.26) | 4.78 | 95.22 |
| Divorced | 79 (0.51) | 0.00 | 100.00 |
| Separated | 371 (2.38) | 5.21 | 94.79 |
| **Total children ever born** | | | |
| Zero | 4117 (26.43) | 3.56 | 96.44 |
| One-three | 6607 (42.42) | 4.65 | 95.35 |
| Four or more | 4851 (31.15) | 3.43 | 96.57 |
| **Employment status** | | | |
| Unemployed | 4831 (31.02) | 2.75 | 97.25 |
| Employed | 10742 (68.98) | 4.54 | 95.46 |
| **Frequency of reading newspaper or magazine** | | | |
| Not at all | 14330 (92.01) | 3.64 | 96.36 |
| Less than once a week | 851 (5.47) | 6.05 | 93.95 |
| At least once a week | 393 (2.52) | 11.93 | 88.07 |
| **Frequency of listening to radio** | | | |
| Not at all | 8653 (55.56) | 2.87 | 97.13 |
| Less than once a week | 3182 (20.43) | 4.80 | 95.20 |
| At least once a week | 3739 (24.01) | 5.86 | 94.14 |
| **Frequency of watching television** | | | |
| Not at all | 11143 (71.55) | 3.59 | 96.41 |
| Less than once a week | 2109 (13.54) | 3.55 | 96.45 |
| At least once a week | 2322 (14.91) | 6.27 | 93.73 |
| **Getting medical help for self: getting permission to go** | | | |
| Not a big problem | 11767 (75.55) | 4.64 | 95.36 |
| Big problem | 3807 (24.45) | 1.95 | 98.05 |
| **Getting medical help for self: getting the money needed for treatment** | | | |
| Not a big problem | 5153 (33.09) | 4.51 | 95.49 |

*(Continued)*

**Table 1.** (Continued)

| Variables | Weighted frequency (%) | Health insurance coverage | |
|---|---|---|---|
| | | Yes (%) | No (%) |
| Big problem | 10421 (66.91) | 3.72 | 96.28 |
| **Getting medical help for self: distance to a health facility** | | | |
| Not a big problem | 8711 (55.93) | 5.51 | 94.49 |
| Big problem | 6863 (44.07) | 2.04 | 97.96 |
| **Getting medical help for self: not wanting to go alone** | | | |
| Not a big problem | 12180 (78.21) | 4.68 | 95.32 |
| Big problem | 3393 (21.79) | 1.48 | 98.52 |
| **Sex of household head** | | | |
| Male | 10930 (70.18) | 4.10 | 95.90 |
| Female | 4644 (29.82) | 3.71 | 96.29 |
| **Wealth index** | | | |
| Poorest | 2738 (17.58) | 1.88 | 98.12 |
| Poorer | 2831 (18.18) | 3.47 | 96.53 |
| Middle | 2954 (18.96) | 3.72 | 96.28 |
| Richer | 3385 (21.74) | 4.12 | 95.88 |
| Richest | 3666 (23.54) | 6.04 | 93.96 |
| **Type of place of residence** | | | |
| Urban | 7163 (46.00) | 5.12 | 94.88 |
| Rural | 8411 (54.00) | 3.01 | 96.99 |
| **Region** | | | |
| Eastern | 3069 (19.71) | 0.91 | 99.09 |
| Northwestern | 2508 (16.10) | 10.29 | 89.71 |
| Northern | 3317 (21.30) | 3.55 | 96.45 |
| Southern | 2900 (18.62) | 1.42 | 98.58 |
| Western | 3780 (24.27) | 4.64 | 95.36 |

campaigns emphasizing the benefits of health insurance, even with a small frequency. However, the association may also reflect underlying socioeconomic factors, as women who listen to the radio may have higher education or income levels, enabling them to better understand and afford health insurance [28]. This finding underscores the importance of media exposure in shaping health insurance enrollment behaviors and suggest that targeted health communication through radio could play a role in increasing coverage [33–35].

In the current study, women who reported having a big problem with the distance to health facilities had higher odds of non-enrollment in health insurance coverage compared to those who did not face such a problem. This finding highlights the critical role of access barriers in shaping health insurance enrollment in Sierra Leone. Women living far from health facilities may face logistical challenges, such as long travel times, limited transportation options, and higher transportation costs, which can hinder their ability to access health services and information about health insurance enrollment [29,33,35]. Additionally, the physical distance may deter women from seeking healthcare altogether, reducing the perceived value of health insurance as they may not see it as a practical or necessary investment if accessing healthcare remains difficult [17]. These logistical and access-related challenges underscore the importance of addressing infrastructural and transportation barriers to improve health insurance coverage, particularly for women in remote areas.

Our results indicate that women living in the Northwestern and Northern regions had lower odds of non-enrollment in health insurance coverage compared to women in the Eastern region of Sierra Leone. This regional difference likely

**Table 2. Factors associated with non-enrollment in health insurance among women in Sierra Leone.**

| Variable | cOR (95% CI) | P-value | aOR (95% CI) | P-value |
|---|---|---|---|---|
| **Age in 5-year groups** | | | | |
| 15-19 | 1 | | – | |
| 20-24 | 0.76 (0.56-1.05) | 0.100 | | |
| 25-29 | 0.62 (0.43-0.88) | 0.008 | | |
| 30-34 | 0.58 (0.35-0.98) | 0.040 | | |
| 35-39 | 0.68 (0.47-1.19) | 0.052 | | |
| 40-44 | 0.75 (0.47-1.19) | 0.223 | | |
| 45-49 | 0.66 (0.38-1.14) | 0.132 | | |
| **Highest educational level** | | | | |
| No education | 1 | | 1 | |
| Primary | 1.40 (0.99-1.97) | 0.056 | 1.33 (0.96,1.83) | 0.077 |
| Secondary | 0.93 (0.68-1.27) | 0.639 | 1.03 (0.78,1.37) | 0.920 |
| Higher | 0.22 (0.14-0.34) | <0.001 | 0.35(0.18,0.64) | 0.001 |
| **Current marital status** | | | | |
| Never in union | 1 | | – | |
| Married | 0.83 (0.67-1.04) | 0.112 | | |
| Cohabiting | 1.38 (0.67-2.82) | 0.381 | | |
| Widow | 0.74 (0.35-1.55) | 0.440 | | |
| Divorced | o.61 (0.28, 1.36) | 0.225 | | |
| Separated | 0.67 (0.35-1.30) | 0.237 | | |
| **Total children ever born** | | | | |
| Zero | 1 | | – | |
| one-three | 0.76 (0.50-1.14) | 0.184 | | |
| Four or more | 1.04 (0.77-1.41) | 0.789 | | |
| **Employment status** | | | | |
| Unemployed | 1 | | 1 | |
| Employed | 0.59 (0.45-0.79) | <0.001 | 0.47 (0.36,0.62) | <0.001 |
| **Frequency of reading newspaper or magazine** | | | | |
| Not at all | 1 | | 1 | |
| Less than once a week | 0.59 (0.36-0.96) | 0.032 | 0.88 (0.44,1.76) | 0.719 |
| At least once a week | 0.28 (0.17-0.46) | <0.001 | 0.59 (0.28,1.25) | 0.172 |
| **Frequency of listening to radio** | | | | |
| Not at all | 1 | | 1 | |
| Less than once a week | 0.59 (0.46-0.75) | <0.001 | 0.71(0.53,0.97) | 0.032 |
| At least once a week | 0.47 (0.31-0.72) | 0.001 | 0.66 (0.40,1.07) | 0.095 |
| **Frequency of watching television** | | | | |
| Not at all | 1 | | – | |
| Less than once a week | 1.01 (0.64-1.59) | 0.966 | | |
| At least once a week | 0.56 (0.30-1.04) | 0.065 | | |
| **Getting medical help for self: getting permission to go** | | | | |
| Not a big problem | 1 | | 1 | |
| Big problem | 2.44 (1.58-3.77) | <0.001 | 1.24 (0.78,1.97) | 0.334 |
| **Getting medical help for self: getting the money needed for treatment** | | | | |
| Not a big problem | 1 | | – | |
| Big problem | 1.22 (0.81-1.84) | 0.340 | | |

*(Continued)*

**Table 2.** (Continued)

| Variable | cOR (95% CI) | P-value | aOR (95% CI) | P-value |
|---|---|---|---|---|
| **Getting medical help for self: distance to a health facility** | | | | |
| Not a big problem | 1 | | 1 | |
| Big problem | 2.80 (1.50-5.23) | 0.001 | 2.21 (1.03,4.75) | 0.041 |
| **Getting medical help for self: not wanting to go alone** | | | | |
| Not a big problem | 1 | | 1 | |
| Big problem | 3.26 (2.14-4.97) | <0.001 | 1.50 (0.86,2.63)8 | 0.146 |
| **Sex of household head** | | | | |
| Male | 1 | | – | |
| Female | 1.11 (0.83-1.47) | 0.476 | | |
| **Wealth index** | | | | |
| Poorest | 1 | | 1 | |
| Poorer | 0.53 (0.33-0.86) | 0.01 | 0.77 (0.49,1.21) | 0.266 |
| Middle | 0.50 (0.24-1.02) | 0.056 | 0.86 (0.42,1.74) | 0.686 |
| Richer | 0.45 (0.21-0.93) | 0.032 | 0.97 (0.35,2.66) | 0.956 |
| Richest | 0.30 (0.13-0.68) | 0.004 | 0.68 [0.23,2.03) | 0.500 |
| **Type of place of residence** | | | | |
| Urban | 1 | | 1 | |
| Rural | 1.74 (0.98-3.08) | 0.058 | 1.59 (0.69,3.62) | 0.269 |
| **Region** | | | | |
| Eastern | 1 | | 1 | |
| Northwestern | 0.08 (0.04-0.16) | <0.001 | 0.07 (0.03,0.14) | <0.001 |
| Northern | 0.25 (0.10-0.61) | 0.002 | 0.28 (0.12,0.66) | 0.004 |
| Southern | 0.64 (0.31-1.30) | 0.216 | 0.57(0.27,1.18) | 0.133 |
| Western | 0.19 (0.08-0.47) | <0.001 | 0.46 (0.17,1.19) | 0.111 |

aOR, adjusted odds ratio; cOR, crude odds ratio; 95% CI, confidence interval; 1, reference category.

reflects broader systemic disparities and contextual factors influencing health insurance enrollment [36]. The Eastern region has historically faced challenges such as limited transport infrastructure, which restricts access to health facilities and health insurance enrollment points, and fewer economic and employment opportunities, which may reduce the ability of women to afford health insurance [37]. Lower levels of female literacy and education in the Eastern region may also hinder women's understanding of the benefits and processes of health insurance enrollment. Additionally, gender-related issues, such as reduced autonomy in decision-making, could further limit women's ability to prioritize health insurance [30,37]. Ethnic and cultural variations may also play a role, influencing trust in formal health systems and preferences for traditional or informal providers. In contrast, women in the Northwestern and Northern regions may benefit from better transport access, higher literacy rates, and greater exposure to health-related information, enabling them to perceive the value of health insurance and access enrollment services more readily. These findings highlight the need to address systemic marginalization and regional inequities by improving infrastructure, education, and economic opportunities in underserved areas like the Eastern region to enhance health insurance coverage.

## Implications for policy and practice

The findings of this study underscore key policy and programmatic priorities to improve health insurance enrollment among women in Sierra Leone. Investing in women's education is crucial, as higher education significantly reduced

non-enrollment, with long-term benefits for health coverage and outcomes. Similarly, economic empowerment through formalizing employment and expanding employer-sponsored insurance schemes, alongside incentives for informal sector workers, could enhance enrollment. Targeted health communication, particularly via radio, offers a cost-effective strategy to raise awareness and bridge knowledge gaps. Addressing geographic barriers through investments in healthcare infrastructure and expanding services in underserved areas is essential to improve access and the perceived value of insurance. Lastly, region-specific interventions, such as subsidies and localized outreach programs, are necessary to tackle disparities and promote equitable health.

### Strengths and Limitations of the study

This study addresses a critical issue regarding access to health insurance as a pathway to achieving universal health coverage and improving women's well-being. While the study benefits from the use of a large, nationally representative SLDHS and standardized data collection procedures, several limitations should be acknowledged.

First, the cross-sectional design of the study limits our ability to establish causal relationships between the explanatory variables and health insurance non-enrollment. While associations were identified, it is not possible to determine causality between variables. For instance, while higher education and employment were associated with lower odds of being uninsured, it is unclear whether these factors directly lead to increased enrollment or whether other unmeasured variables play a role. Future research could address this limitation by employing longitudinal study designs to better understand how changes in socioeconomic and demographic factors influence health insurance enrollment over time.

Second, the SLDHS dataset, while robust, does not provide information on the type of health insurance schemes (e.g., public or private), limiting our ability to analyze whether specific types of insurance are more accessible or effective for women. Future studies could explore this distinction to provide more targeted recommendations for improving health insurance coverage.

Finally, while the study highlights key socioeconomic and geographic factors influencing health insurance non-enrollment, it does not fully account for the cultural and social factors that may shape women's decisions to enroll in health insurance. These factors may include perceptions of health insurance, trust in healthcare systems, and traditional norms, which are difficult to capture in a cross-sectional survey. Qualitative or mixed-methods research could provide deeper insights into these contextual factors and help design culturally appropriate interventions to promote health insurance enrollment.

### Conclusion

This study reveals a very low uptake of health insurance among women in Sierra Leone, underscoring a notable barrier to achieving universal health coverage. Women with higher education, employment, and access to health communication via radio were less likely to be uninsured, while those facing geographic barriers, such as distance to health facilities, and those in the Eastern region were more likely to be uninsured. These findings suggest that interventions should prioritize improving education, employment opportunities, healthcare accessibility, and targeted health communication, particularly for women in underserved regions like the Eastern region. Future research should focus on exploring cultural and financial barriers to health insurance enrollment to inform more inclusive and effective policies.

### Author contributions

**Conceptualization:** Augustus Osborne, Mainprice Akuoko Essuman, Peter Bai James, Camilla Bangura, Bright Opoku Ahinkorah.

**Data curation:** Mainprice Akuoko Essuman.

**Formal analysis:** Mainprice Akuoko Essuman.

**Methodology:** Augustus Osborne, Mainprice Akuoko Essuman, Richard Gyan Aboagye, Bright Opoku Ahinkorah.

**Supervision:** Augustus Osborne, Bright Opoku Ahinkorah.

**Validation:** Bright Opoku Ahinkorah.

**Writing – original draft:** Augustus Osborne, Mainprice Akuoko Essuman, Peter Bai James, Camilla Bangura, Richard Gyan Aboagye, Comfort Z. Olorunsaiye, Abdul Aziz Seidu, Bright Opoku Ahinkorah.

**Writing – review & editing:** Augustus Osborne, Mainprice Akuoko Essuman, Peter Bai James, Camilla Bangura, Richard Gyan Aboagye, Comfort Z. Olorunsaiye, Abdul Aziz Seidu, Bright Opoku Ahinkorah.

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
