## [Decision Letter · Decision Letter 0]

Dear Dr. Osborne,

Thank you for submitting your manuscript to PLOS ONE. After careful consideration, we feel that it has merit but does not fully meet PLOS ONE’s publication criteria as it currently stands. Therefore, we invite you to submit a revised version of the manuscript that addresses the points raised during the review process.

**ACADEMIC EDITOR:** The reviewers have highlighted several issues that the authors need to carefully address. It is essential to revise the manuscript accordingly, making necessary corrections or providing a well-reasoned rebuttal for each point, as appropriate.

We look forward to receiving your revised manuscript.

Kind regards,

Emmanuel O Adewuyi, BPharm, MPH, PhD

Academic Editor

PLOS ONE

Additional Editor Comments (if provided):

Reviewers' comments:

Reviewer's Responses to Questions

**Comments to the Author**

1. Is the manuscript technically sound, and do the data support the conclusions?

Reviewer #1: Partly

Reviewer #2: Yes

Reviewer #3: Yes

2. Has the statistical analysis been performed appropriately and rigorously?

Reviewer #1: No

Reviewer #2: Yes

Reviewer #3: Yes

3. Have the authors made all data underlying the findings in their manuscript fully available?

Reviewer #1: Yes

Reviewer #2: Yes

Reviewer #3: Yes

4. Is the manuscript presented in an intelligible fashion and written in standard English?

Reviewer #1: Yes

Reviewer #2: Yes

Reviewer #3: Yes

Reviewer #1: Abstract: Introduction: The rationale of the study is poorly explained. Need to ensure the reason for the current study.  It is important to detail why women become the population group. If there is not enough space, it is alright to add that in the main text. 

Methods:  'Multivariable binary logistic regression analysis'- use standard statistical terminology. There is no reason to invent new terms such as this. I believe you have used 'multiple logistic regression'. 

Result: Currently odds ratio is hard to understand double negative direction. If non-enrolment is the outcome of interest- try switching the reference category of each variable so it is easier to understand the message. You have done that in the second half of result. 

Conclusion: 'Again, efforts should be made to eliminate financial barriers that prevent women from being able to afford health insurance enrolment.'- The significant variables do not explain or examine financial barrier in your study.  In general men and women both who face financial barrier are at risk of not having health insurance. However, that is not what you have investigated in this study.  Please use your current results only to recommend the interventions.

Main text: Introduction : Line 74-79- This paragraph does not add value to your study. The second paragraph almost conveys the same message. I would expect to read some challenges in health financing globally, leading to health insurance as one of the measures to help people access health care services.  It is also essential to establish the global, regional and national context, and then lead the literature review to the local context. The authors have included the later. There is good synthesis of the local context. However, it lacks a broader context. Line 101-102- I think health insurance goes beyond just reducing maternal mortality, unless it is restricted in the country. Please check if the guideline says that only maternal health services are made available or it also extends to universal health care/primary health care. 

Line 124-125: 'However, health insurance coverage and impact in125 Sierra Leone is very low, especially among women'. Please add data on overall coverage, and brief comparison of men and women to support your claim. 

Line 125-126- The authors also need to establish the literature gap. Why is there a need to conduct this study? I am sure authors have gone through numerous articles and identified the gap. Please summarize briefly. 

Methods

Line 133-145: Please mention which name of the data set. DHS study is quite unique that it has multiple datasets and each dataset has its own strength. It is essential to ensure transparency. I also believe the authors have used 'the Guide to DHS statistics' which clearly outlines the dataset that needs to be analysed for certain variables. 

Variables: I believe this can be restructured a bit better. Please use sub-heading- 'Outcome variable' and 'Independent/exploratory variables' so it is easier. It is also essential to provide references to the relevant studies to support how the authors categorised the variables.

Statistical analysis- as mentioned earlier- it is essential to use standard statistical method which authors have done. Please use the same term consistently.

Line 185-189: This can be shortened . 'Weighted analysis was conducted using XX command in Stata. 

Results: Table 1. looks very busy. It is needless to report both weighted and unweighted frequency distribution. I suggest to report weighted column only.  In the same table though, the health insurance part does not have frequency distribution. Since the enrolment rate is very low, it is essential to see the frequency distribution too. I believe the authors proceed with binary logistic regression without performing chi-square test. If that is the case, the reason has to be explained in  analysis section. 

Line 207- 'Predictor' is a very strong word. With a cross sectional study such as DHS, we can not ascertain predictors. The most we can do is to explore factors associated. Please revise the heading accordingly. Similar to what I mentioned earlier, the authors need to present result in one direction. 'Non-enrolment' is already a negative outcome. Double negative in results has made it difficult to understand. 

Table 2- Model 1 and Model 2; This is misleading. Please just write: Crude Odds Ratio ( 95% Confidence Interval), and Adjusted Odds Ratio ( 95% Confidence Interval). Model 1 is not actually one model. It is run separately for each independent variable. P-value  should be: p-value cOR: it is alright to mention: odds ratio and Adjusted odds ratio to make it more readable. 

Table 1- is very long.  I would suggest- keep the entire table in supplementary file.  And, only the significant results  in the main file. It will improve the readability of the manuscript. Those who are interested in detailed results can always refer to supplementary files.  

Please re run multiple logistic regression. If you have screened the association using binary logistic regression, you need to include only significant variables in subsequent multiple logistic regression. For instance, 'total number of children ever born'- is not significant in binary regression- but has been included in multiple logistic regression.  

Line 208- onwards needs to change after the analysis. List the variables which were significant in binary logistic regression. Then explain those which were significant in multiple logistic regression. 

  Table 2: Current marital status  - Please re-categorise the variable. There '1' observation in one category, yet it is included as is. There is no reason to use  ( ) and [ ] to indicate the odds ratio. It is acceptable to use (  ) as there is always a column heading in the table. 

Discussion: This is extremely long discussion section. This needs to be shortened significantly and just focus on your aim. This might change after re-analysis. Please edit/rewrite as appropriate after you have the new results but ensure that it is not long. 

Reference: Ref #1,12, 14,16 and more- there are significant issues. Please ensure it is according to the journal guidelines.

Reviewer #2: Dear authors,

Thank you for reviewing the study entitled “ Determinants of Non-enrollment on Health Insurance among Women in Sierra Leone: a cross-sectional analysis of the 2019 Sierra Leone Demographic Health Survey." This topic is important; however, the study has some shortcomings. I have some comments that I hope the authors will take into account in improving the manuscript.

Abstract: I find that your summary is very long.

Introduction:

Your introduction is well-structured and contains all the necessary information about health insurance. I congratulate you on that. However, some remarks have been raised:

- Support your study with other results related to the effect of financial inaccessibility due to the lack of universal health coverage on the sexual and reproductive health of women of childbearing age ;

Include references for the information presented. Exemple : Line 101: include references for the information presented on Sierra Leone.

Materials and Methods:

Your method is excellently designed. Well done.

Results:

- I invite you to present the main results associated with the lack of enrollment in health insurance according to various variables related to the sociodemographic and socioeconomic characteristics of women in Sierra Leone.

- Please review the percentages in the tables. If they are above or below 100%. For example, in Table 1: Current marital status, Married.

Discussion

You discussed your results well. However, it will be stronger if supported by other studies in similar contexts.

Reviewer #3: METHODOLOGY

Strengths:

Clear Sampling Design: The study uses a well-organized sampling method called stratified two-stage cluster sampling, which is reliable for large surveys. The process of selecting households and enumeration areas is clearly explained, making the sampling process transparent.

Detailed Variable Description: The variables used in the study are well-defined, covering a wide range of factors like sociodemographic and health-related details. This strengthens the analysis, particularly when looking at health insurance non-enrollment.

Thorough Statistical Analysis: The study uses both descriptive and inferential statistics, including logistic regression, which is appropriate for its goals. The report also checks for issues like multicollinearity, ensuring that the model used is accurate.

Following Guidelines: The study follows the STROBE guidelines, which adds to its credibility by ensuring methodological rigor.

Areas for Improvement:

Ethical Considerations: While the study mentions informed consent, more details on ethical aspects like data confidentiality and protection of participants would improve transparency. Information on institutional ethical approval should also be added.

Clarifying Explanatory Variables: Some variable categories, like wealth or marital status, need more explanation. For example, defining what makes someone "poorest" or "richest" would add clarity.

Handling Missing Data: The study uses methods to adjust for sampling and non-response but could provide more information on how it handled missing data, a common issue in large surveys.

Dependent Variable Coding: The recoding of health insurance enrollment (1 = no, 0 = yes) could be confusing. Explaining why this approach was used would clarify things for readers.

Presentation of Results: While the study uses odds ratios (aOR) appropriately, it would help to mention how they checked that the model was a good fit, using tests like goodness-of-fit.

Justifying Variable Selection: The study selects explanatory variables based on past research, but it could elaborate on why specific variables (e.g., region, employment) were chosen and how these have been linked to health insurance in previous studies.

Analysis Software: Stata 14 is a good software choice, but it might be useful to explain why this version was used, as newer versions offer improved features.

Overall: The methodology section is detailed and could be replicated. However, providing more information on ethics, defining variables more clearly, and explaining some choices (like recoding) would make it even stronger.

RESULTS

Clarity and Structure: The results section is repetitive in some parts, especially when discussing percentages of health insurance non-enrollment. Simplifying the language and making it more concise would improve readability. Also, ensure that all figures and tables are labeled correctly and placed in the right sections of the text.

Descriptions of Tables: The transition from figures to tables can be smoother. For example, before introducing Table 1, provide a brief explanation of what the table will show, such as a breakdown of sociodemographic factors.

Interpreting the Data: When discussing variations in non-enrollment, be specific about which variables show significant differences (e.g., age, education). Highlight key findings so readers don’t have to sift through all the data themselves.

Predictors Section: The analysis of predictors is comprehensive, but the significant findings should be emphasized more clearly. For example, when discussing differences between regions, mention the exact odds ratios to give readers a clearer picture.

Terminology and Precision: Avoid vague language. For example, instead of saying, “There are variations in the distribution of explanatory variables,” specify which variables vary, like age or education.

Contextualizing Findings: Provide some insight into why certain groups are more likely to be uninsured. Although this is mainly for the discussion, a brief mention here would prepare readers for the interpretation later.

DISCUSSION

Strengths:

Identification of Key Issues: The discussion identifies important factors that influence health insurance enrollment, such as lack of awareness, financial barriers, and differences between regions.

Context with Other Studies: Similar findings from other studies are used to contextualize the results within the broader literature, which is a good approach.

Policy Implications: The discussion suggests practical policy solutions, such as outreach programs and financial incentives, to increase health insurance coverage.

Areas for Improvement:

Repetitiveness: Some points, like financial barriers and lack of awareness, are repeated. These can be consolidated to improve the flow of the discussion.

Depth of Analysis: Some explanations need more evidence. For example, linking low media exposure to income levels feels speculative. Explore cultural and social factors in more depth, like how local beliefs affect health insurance enrollment.

Inconsistencies in Findings: Some findings seem contradictory, such as why women with higher education enroll less often. More explanation is needed to clarify these points.

Discussion Structure: Group related factors under clear subheadings (e.g., "Socioeconomic Barriers") to make the discussion easier to follow.

Data Limitations: The limitations section could elaborate more on the study’s cross-sectional design and how future research could address this.

Policy Recommendations: The policy suggestions could be more detailed. For example, explain how financial incentives or mobile technology could help increase enrollment in rural areas.

Conclusion: The conclusion could summarize key findings more directly and offer a future research agenda to address the study's limitations.

**Do you want your identity to be public for this peer review?** For information about this choice, including consent withdrawal, please see our Privacy Policy

Reviewer #1: No

Reviewer #2: No

Reviewer #3: No

---

## [Author Response · Author response to Decision Letter 1]

4 Mar 2025

The Editor

PLOS ONE

1st March 2025

Ref: PONE-D-24-25104

Title: Determinants of Non-enrollment on Health Insurance among Women in Sierra Leone: a cross-sectional analysis of the 2019 Sierra Leone Demographic Health Survey.

Response to Reviewers' comments

Dear Sir/Madam,

We want to express our sincere thanks for painstakingly reviewing our manuscript and providing valuable comments and suggestions. Please see our point-by-point response to the reviewers' comments and suggestions. Revisions are highlighted with track changes in the revised manuscript.

ACADEMIC EDITOR:

The reviewers have highlighted several issues that the authors need to carefully address. It is essential to revise the manuscript accordingly, making necessary corrections or providing a well-reasoned rebuttal for each point, as appropriate.

Response: Thank you. We have carefully addressed all reviewer comments to help strengthen our manuscript.

Reviewer #1: Abstract: Introduction: The rationale of the study is poorly explained. Need to ensure the reason for the current study. It is important to detail why women become the population group. If there is not enough space, it is alright to add that in the main text.

Response: Health insurance enrollment is a vital component of achieving universal health coverage (UHC) and improving access to essential healthcare services. However, in Sierra Leone, enrollment remains persistently low, posing a significant public health challenge. Women aged 15–49 represent a critical population for this issue due to their unique healthcare needs, particularly related to reproductive health, pregnancy, and childcare, as well as their central role in household and community health. Despite their importance, women face significant barriers to health insurance enrollment, including financial constraints, gender inequalities, and sociocultural factors. This study examines the factors associated with non-enrollment of health insurance among women in this demographic, aiming to inform targeted interventions to improve coverage and advance UHC in Sierra Leone.

Methods: 'Multivariable binary logistic regression analysis'- use standard statistical terminology. There is no reason to invent new terms such as this. I believe you have used 'multiple logistic regression'.

Response: Multiple logistic regression analysis was used to examine the factors associated with health insurance non-enrollment among the women.

Result: Currently odds ratio is hard to understand double negative direction. If non-enrolment is the outcome of interest- try switching the reference category of each variable so it is easier to understand the message. You have done that in the second half of result.

Response: The proportion of health insurance non-enrollment among the women was 96.0%. Women with higher education were significantly less likely to be uninsured (aOR: 0.35, 95% CI: 0.18-0.64, p=0.001) compared to those with no education, indicating that higher education was protective against non-enrollment. Employment also reduced the odds of being uninsured (aOR: 0.47, 95% CI: 0.36-0.62, p<0.001) compared to unemployment. Listening to the radio less than once a week was associated with lower odds of being uninsured (aOR: 0.71, 95% CI: 0.53-0.97, p=0.032) compared to not listening at all. On the other hand, women who reported distance to a health facility as a big problem were more likely to be uninsured (aOR: 2.21, 95% CI: 1.03-4.75, p=0.041) compared to those who did not consider it a problem. Regionally, women living in the Northwestern (aOR: 0.07, 95% CI: 0.03-0.14, p<0.001) and Northern regions (aOR: 0.28, 95% CI: 0.12-0.66, p=0.004) were significantly less likely to be uninsured compared to those in the Eastern region.

Conclusion: 'Again, efforts should be made to eliminate financial barriers that prevent women from being able to afford health insurance enrolment.'- The significant variables do not explain or examine financial barrier in your study. In general men and women both who face financial barrier are at risk of not having health insurance. However, that is not what you have investigated in this study. Please use your current results only to recommend the interventions.

Response: Health insurance non-enrollment remains very high among women in Sierra Leone.Higher education and employment were protective against non-enrollment, underscoring the need to address socioeconomic barriers and promote education and economic empowerment to increase coverage. The association between listening to the radio and reduced odds of non-enrollment suggests that leveraging mass media campaigns could be an effective strategy to raise awareness and promote health insurance uptake. Conversely, the increased likelihood of non-enrollment among women facing distance-related challenges to healthcare access highlights the importance of improving geographic accessibility to health services. Regional disparities further emphasize the need for targeted interventions, such as community-based outreach programs to educate women on the benefits of health insurance.

Main text: Introduction : Line 74-79- This paragraph does not add value to your study. The second paragraph almost conveys the same message. I would expect to read some challenges in health financing globally, leading to health insurance as one of the measures to help people access health care services. It is also essential to establish the global, regional and national context, and then lead the literature review to the local context. The authors have included the later. There is good synthesis of the local context. However, it lacks a broader context.

Response: In recent years, global health policy has increasingly focused on achieving universal health coverage (UHC) as a means of ensuring equitable access to healthcare services and protecting individuals from catastrophic out-of-pocket (OOP) healthcare expenses [1-3]. Health financing remains a significant challenge worldwide, particularly in low- and middle-income countries (LMICs), where high OOP expenditures often hinder access to essential healthcare services and push millions of households into poverty each year [4]. To address these challenges, health insurance has emerged as a critical mechanism to improve access to healthcare, reduce financial barriers, and promote equity in health systems. However, despite its potential, health insurance coverage remains uneven globally, with significant gaps in enrollment, particularly among vulnerable populations such as women, children, and those in low-income settings [5].

In sub-Saharan Africa (SSA), health insurance policies have been implemented over the past two decades as part of broader efforts to achieve UHC and meet the Sustainable Development Goals (SDGs), particularly SDG 3.8 (achieving UHC) and SDG 3.1 (reducing maternal mortality) [6]. Countries such as Ghana, Rwanda, Kenya, Nigeria, and South Africa have introduced various health insurance programmes to improve healthcare accessibility and reduce financial strain on their populations [7]. These programmes have targeted both formal sector employees and informal sector workers, with some extending coverage to entire communities. Despite these efforts, enrollment rates in many SSA countries remain low, with significant disparities in coverage between urban and rural populations, and between men and women [8]. Women, particularly those in their reproductive years, face unique barriers to enrollment, including economic constraints, limited awareness, and sociocultural factors that restrict their access to healthcare services [9].

Sierra Leone, a low-income country in West Africa with a population of approximately 7.9 million, exemplifies these challenges. Women, who constitute 51% of the population, bear a disproportionate burden of healthcare needs, particularly related to maternal and reproductive health [10]. The country continues to face a high maternal mortality ratio, recorded at 717 deaths per 100,000 live births in 2019, and a high adolescent birth rate of 102 births per 1,000 women aged 15–19 [11]. Despite the introduction of health insurance policies in Sierra Leone, enrollment remains persistently low, particularly among women, posing a significant barrier to achieving UHC and improving maternal health outcomes. Understanding the factors that contribute to low health insurance enrollment among women is critical for addressing these disparities and advancing health equity in Sierra Leone.

Line 101-102- I think health insurance goes beyond just reducing maternal mortality, unless it is restricted in the country. Please check if the guideline says that only maternal health services are made available or it also extends to universal health care/primary health care.

Response: Health insurance is a vital strategy for ensuring universal access to affordable and quality healthcare, particularly for vulnerable groups like women and children [12]. While maternal health services are often prioritized in low-income countries such as Sierra Leone due to high maternal mortality rates, health insurance schemes typically aim to provide broader coverage, including universal and primary health care services [12]. These services encompass preventative care, treatment for various diseases, and essential health interventions. In Sierra Leone, health insurance supports the broader goal of universal health coverage, making it crucial to address barriers to enrollment among women to improve overall health outcomes and advance equity in healthcare access [12].

Line 124-125: 'However, health insurance coverage and impact in Sierra Leone is very low, especially among women'. Please add data on overall coverage, and brief comparison of men and women to support your claim.

Response: Previous research on health insurance in Sierra Leone has focused on various aspects, such as the willingness to pay for health insurance in the informal sector [15] and population characteristics influencing the benefits basket of the national social health insurance scheme [16]. Health insurance has been shown to improve health outcomes and reduce poverty among women and their families in Sierra Leone [12]. It reduces out-of-pocket payments for healthcare, increases the utilisation of preventive and curative services, and enhances the quality and efficiency of healthcare delivery [17]. However, health insurance coverage in Sierra Leone remains alarmingly low. According to the 2019 Sierra Leone Demographic and Health Survey (SLDHS), only 4% of women and 3% of men are enrolled in health insurance schemes, highlighting a critical disparity in access to financial protection for healthcare [11]. This low coverage reflects significant barriers to enrollment, particularly for women, who face unique social, economic, and cultural challenges.

Line 125-126- The authors also need to establish the literature gap. Why is there a need to conduct this study? I am sure authors have gone through numerous articles and identified the gap. Please summarize briefly.

Response: Despite the evidence on the potential benefits of health insurance, there is limited research examining the factors contributing to the low enrollment of women in health insurance schemes in Sierra Leone. Existing studies have largely focused on willingness to pay or general population characteristics without delving into the gender-specific barriers that women face [15, 16]. Furthermore, while health insurance is widely recognized as a pro-poor measure, there is a lack of comprehensive analysis on its gendered impact and the specific challenges women encounter in accessing health insurance in low-income settings like Sierra Leone. This study addresses this gap by exploring the factors influencing the low enrollment of women in health insurance using the 2019 SLDHS. The findings will contribute to the existing literature by providing gender-specific insights and evidence-based recommendations to improve health insurance coverage and, ultimately, the health and well-being of women and their families in Sierra Leone.

Methods

Line 133-145: Please mention which name of the data set. DHS study is quite unique that it has multiple datasets and each dataset has its own strength. It is essential to ensure transparency. I also believe the authors have used 'the Guide to DHS statistics' which clearly outlines the dataset that needs to be analysed for certain variables.

Response: The 2019 SLDHS used for this study is part of a series of periodic cross-sectional surveys designed to collect data on demographic, health, and nutritional factors among non-elderly adults and children. The most recent 2019 SLDHS was conducted over four months, from May 2019 to August 2019 [18]. The survey employed a stratified, two-stage cluster sampling design. In the first stage, 578 enumeration areas (EAs) were selected, comprising 214 urban and 364 rural regions. In the second stage, 24 households were systematically selected from each EA, resulting in a total sample size of 13,872 households [18]. For this study, the Individual Recode (IR) dataset from the 2019 SLDHS was used. This dataset is specifically designed to provide detailed information on women aged 15 to 49, including their sociodemographic characteristics, health indicators, and access to healthcare services. The IR dataset was selected because it is the most appropriate for analyzing factors influencing women’s health insurance enrollment. The choice of variables and analysis follows the recommendations outlined in the "Guide to DHS Statistics," which provides a comprehensive framework for identifying the relevant datasets and variables for specific research objectives. The survey collected sociodemographic information from respondents using interviewer-administered questionnaires. A total of 15,574 women aged 15 to 49 were interviewed, and all participants provided written informed consent. Detailed information regarding the sampling technique and survey methodology can be found in the final DHS report [18].

Variables: I believe this can be restructured a bit better. Please use sub-heading- 'Outcome variable' and 'Independent/exploratory variables' so it is easier. It is also essential to provide references to the relevant studies to support how the authors categorised the variables.

Response: Outcome variable

The dependent variable in this study was health insurance coverage. This was derived from the question, 'Does any health insurance cover you?' It was coded 1 = “Yes” and 0 = “No”. However, to derive health insurance non-enrollment, we recoded the response options as "1=no" and "0=yes" in the final analysis, where "no" referred to women who were covered by health insurance and "yes" to those who were not enrolled on health insurance.

Explanatory variables

We included sixteen explanatory variables in the study. These variables consisted of age classified in a 5–year grouping and categorised as 15–19 = 1, 20–24 = 2, 25-29=3, 30-34=4, 35-39=5, 40-44=6, 45-49=7. Levels of education were captioned as no education = 1, primary = 2, secondary = 3, and higher education = 4. Employment status was categorised as unemployed= 0 and employed =1. Wealth status was categorized as poorest = 1, poorer = 2, middle = 3, richer = 4, and richest = 5. The sex of household head was categorised as male=1 and female=2. Marital status was categorized as never in union =1, married =2, cohabitating =3, widowed =4, divorced =5, and separated =6. The total number of children born was categorised as zero birth=0, one to three births=2 and Four or more births=3. Challenges with getting medical help for self (i.e., getting permission to go, getting the money needed for treatment, and distance to health facility) were categorised as a big problem =1 and not a big problem=2. The frequency of reading newspapers, listening to radio, and viewing television was not at all = 1, less than once a week = 2, and at least once a week = 3. Place of residence was categorised as Urban=1 and Rural=2. Region captured as Eastern=1, Northwestern=2 Northern=3, Southern=4, and Western=5. The selection of the explanatory variables was guided by their presence in the DHS datasets and their association with health insurance coverage, as indicated by prior res

---

## [Decision Letter · Decision Letter 1]

Dear Dr. Osborne,

Thank you for submitting your manuscript to PLOS ONE. After careful consideration, we feel that it has merit but does not fully meet PLOS ONE’s publication criteria as it currently stands. Therefore, we invite you to submit a revised version of the manuscript that addresses the points raised during the review process.

**ACADEMIC EDITOR:** There are a few minor comments from the reviewers. I now invite the authors to respond to the comments and make corrections as necessary. I suggest that the authors change 'determinants' in their title to 'factors associated with ...,' as 'determinants' can imply causation, which is not the case in the current study. Additionally, the authors should thoroughly proofread their work (line by line) and fact-check every statement to ensure smooth progress without delay. I look forward to receiving the revised manuscript.

We look forward to receiving your revised manuscript.

Kind regards,

Emmanuel O Adewuyi, BPharm, MPH, PhD

Academic Editor

PLOS ONE

Journal Requirements:

Reviewers' comments:

Reviewer's Responses to Questions

**Comments to the Author**

Reviewer #1: (No Response)

Reviewer #2: All comments have been addressed

2. Is the manuscript technically sound, and do the data support the conclusions?

Reviewer #1: Partly

Reviewer #2: Yes

3. Has the statistical analysis been performed appropriately and rigorously?

Reviewer #1: Yes

Reviewer #2: Yes

4. Have the authors made all data underlying the findings in their manuscript fully available?

Reviewer #1: No

Reviewer #2: Yes

5. Is the manuscript presented in an intelligible fashion and written in standard English?

Reviewer #1: Yes

Reviewer #2: Yes

Reviewer #1: The authors have responded the majority of my suggestion. I did not review the last version discussion as the results were likely to change. I have provided some minor suggestions for authors to address. 

Abstract: The authors have used aOR with p- value. Since 95% CI is reported, it is needless to report p-values

Introduction, Line 198:  only 4% of women and 3% of men are enrolled in health insurance schemes ... was added in response to earlier feedback ' 'However, health insurance coverage and impact in Sierra Leone is very low, especially among women'. Please add data on overall coverage, and brief comparison of men and women to support your claim.'  If we look at the revised version, women have higher enrolment than men. How do you justify the research gap is still an issue. 

Explanatory variables: Great to see a better explanation of the explanatory variables. I advise authors to  structure this in paragraphs, not in numbering that is usually done in powerpoints. 

Results sections referring to Table 1;  "Table 1- is very long." in response to these authors have suggested the unweighted frequencies are removed which is good. The table is 4 page long in its current form which is very long for any publication. I would have kept the long table in a supplementary file, and the rest in main file. It still ensures transparency as it comes with the main article. At times the Authors have to be kind to readers as well. Making articles very long is extremely unhelpful.  I will leave it to the academic editor to make a decision.

Results section referring to Table 2: The effect sizes are written (aOR: 0.35, 95% CI: 0.18-0.64, p=0.001). Please remove p-value throughout as that authors have written 95% CI. which is the better measure to show the association. 

Discussion: Line 344-353: Those who listen to radio less than a week had lower odds compared but not those who did so once a week. I think this result is very hard to justify. You argument that listening to the radio as protective for health insurance enrolment is plausible but your result does not support. May be try merging the categories 'not at all' vs 'listens to radio (less than a week or at least once a week). If you get a result where listening to the radio comes with less likelihood of non-enrollment, your current discussion can support. Alternatively, the discussion has to reflect what you have in table 2. 

  Line 354-362: Getting medical help for self: distance to a health facility: is important and same old issue. Multiple 'Tyranny of distance' articles have been written. It is not so much about 'quality' and 'trust' it is more about 'access issue' due to various logistical issues. 

Line: 363: Regional difference is very important for any country. There may be inherent issues due to systemic marginalization of certain region. I would have thought authors provide some contextual information . There is opportunity to make this paragraph really important. Think about transport access, economic and employability opportunities, education, female literacy, gender issues, ethnic variations in the regions that are prone to non-enrolment. There is very important for authors to explore. 

Conclusion: Public health implications have been explained in in earlier section. I would suggest authors make the conclusion really short. For example in one paragraph: highlight key points: Very low uptake, and which groups should be the focus of interventions and end with one future research.

Reviewer #2: Dear authors,

Thank you for studying the article entitled “Determinants of Non-enrollment in Health Insurance among Women in Sierra Leone: a cross-sectional analysis of the 2019 Sierra Leone Demographic Health Survey.” This topic is significant, but the study has some shortcomings. I would like to share a few comments that could help improve the manuscript.

Abstract: I find that the conclusion of your abstract is very long.

Introduction:

Your introduction is well-structured and contains all the necessary information about health insurance. I congratulate you on that. However, some remarks have been made:

• Include references for the information presented. For example: line 113: include references for the information presented on the education level of women in Sierra Leone.

Materials and Methods:

Your method is very well written. Well done.

Results:

Your results have been well presented, emphasizing the main findings related to non-enrollment in health insurance according to the various sociodemographic and socioeconomic characteristics of women in Sierra Leone. Great job.

Discussion:

Your discussion identifies the important factors that influence enrollment in health insurance, particularly the lack of awareness, financial barriers, and differences between regions. Well done.

Recommendations:

Avoid repetitions and try to be concise, especially regarding awareness, policy implications, and strategic recommendations.

Conclusion: Well written. Great job.

**Do you want your identity to be public for this peer review?** For information about this choice, including consent withdrawal, please see our Privacy Policy

Reviewer #1: No

Reviewer #2: No

---

## [Author Response · Author response to Decision Letter 2]

2 Apr 2025

The Editor

PLOS ONE

1st April 2025

Ref: PONE-D-24-25104

Title: Determinants of Non-enrollment on Health Insurance among Women in Sierra Leone: a cross-sectional analysis of the 2019 Sierra Leone Demographic Health Survey.

Response to Reviewers' comments

Dear Sir/Madam,

We want to express our sincere thanks for painstakingly reviewing our manuscript and providing valuable comments and suggestions. Please see our point-by-point response to the reviewers' comments and suggestions. Revisions are highlighted with track changes in the revised manuscript.

ACADEMIC EDITOR:

There are a few minor comments from the reviewers. I now invite the authors to respond to the comments and make corrections as necessary. I suggest that the authors change 'determinants' in their title to 'factors associated with ...,' as 'determinants' can imply causation, which is not the case in the current study. Additionally, the authors should thoroughly proofread their work (line by line) and fact-check every statement to ensure smooth progress without delay. I look forward to receiving the revised manuscript.

Response: Thank you. We have now changed the title to factors associated with and have have thoroughly proofread the article line by line to ensure smooth progress without delay.

Reviewer #1: The authors have responded the majority of my suggestion. I did not review the last version discussion as the results were likely to change. I have provided some minor suggestions for authors to address.

Response: Thank you.

Abstract: The authors have used aOR with p- value. Since 95% CI is reported, it is needless to report p-values

Response: Thank you. We have now deleted the p values in the abstract.

Introduction, Line 198: only 4% of women and 3% of men are enrolled in health insurance schemes ... was added in response to earlier feedback ' 'However, health insurance coverage and impact in Sierra Leone is very low, especially among women'. Please add data on overall coverage, and brief comparison of men and women to support your claim.' If we look at the revised version, women have higher enrolment than men. How do you justify the research gap is still an issue.

Response: Health insurance coverage in Sierra Leone remains alarmingly low, with only 4% of women and 3% of men enrolled in health insurance schemes, according to the 2019 Sierra Leone Demographic and Health Survey (SLDHS) [11]. Overall, this translates to just 3.5% of the population having access to health insurance, underscoring the critical lack of financial protection for healthcare [11]. While women have a slightly higher enrollment rate than men, this marginal difference does not negate the significant barriers to enrollment faced by both genders, particularly women. Women in Sierra Leone encounter unique social, economic, and cultural challenges, such as lower income levels, reduced decision-making power, and limited access to information about health insurance schemes [12]. These factors exacerbate disparities in healthcare access and financial security, making the low overall coverage and the gendered barriers to enrollment a pressing issue for policymakers and researchers alike.

Explanatory variables: Great to see a better explanation of the explanatory variables. I advise authors to structure this in paragraphs, not in numbering that is usually done in powerpoints.

Response: We included sixteen explanatory variables in the study, selected based on their availability in the DHS datasets and their established association with health insurance coverage in prior research [5, 21–23]. These variables were categorized into distinct groups to facilitate analysis.

Age was classified into seven 5-year groups: 15–19, 20–24, 25–29, 30–34, 35–39, 40–44, and 45–49. Each group was coded numerically from 1 to 7, respectively. Education level was categorized based on the highest level of formal education attained by the respondent: no education (1), primary (2), secondary (3), and higher education (4). Employment status was determined based on respondents’ self-reported engagement in income-generating activities during the survey period, categorized as unemployed (0) and employed (1).

The wealth index, a composite measure of a household’s cumulative living standard calculated by the DHS using principal component analysis, was divided into quintiles: poorest (1), poorer (2), middle (3), richer (4), and richest (5). This index considers household ownership of selected assets, housing construction materials, and access to water and sanitation facilities. Sex of household head was categorized as male (1) or female (2), while marital status was classified into six categories: never in union (1), married (2), cohabiting (3), widowed (4), divorced (5), and separated (6), based on respondents’ self-reported marital situation.

The total number of children ever born was categorized into three groups: zero births (0), one to three births (2), and four or more births (3). We also assessed challenges with getting medical help for oneself across three dimensions: getting permission to go, obtaining the money needed for treatment, and distance to the health facility. Each dimension was categorized as a big problem (1) or not a big problem (2), based on respondents’ perceptions.

Frequency of media exposure was measured by how often respondents read newspapers or magazines, listened to the radio, or watched television. This was categorized as not at all (1), less than once a week (2), and at least once a week (3). The place of residence was classified as urban (1) or rural (2), based on the DHS classification of the respondent’s living area. Lastly, region was categorized into five geographic areas of Sierra Leone: Eastern (1), Northwestern (2), Northern (3), Southern (4), and Western (5).

Results sections referring to Table 1; "Table 1- is very long." in response to these authors have suggested the unweighted frequencies are removed which is good. The table is 4 page long in its current form which is very long for any publication. I would have kept the long table in a supplementary file, and the rest in main file. It still ensures transparency as it comes with the main article. At times the Authors have to be kind to readers as well. Making articles very long is extremely unhelpful. I will leave it to the academic editor to make a decision.

Response: Thank you for your feedback regarding the length of Table 1. We understand that the current formatting makes the table appear long in the manuscript. However, the length is primarily due to the double-spacing and layout requirements for submission. Once the manuscript is formatted for publication by PLOS ONE, the table will fit within a single page or at most two pages, making it much more concise and reader-friendly.

We believe that keeping Table 1 within the main manuscript is important for transparency and accessibility, as it provides critical details on the distribution of health insurance coverage across various sociodemographic and socioeconomic characteristics. Including this table in the main file ensures that readers have immediate access to this information without needing to refer to supplementary files, which can sometimes disrupt the flow of reading.

Results section referring to Table 2: The effect sizes are written (aOR: 0.35, 95% CI: 0.18-0.64, p=0.001). Please remove p-value throughout as that authors have written 95% CI. which is the better measure to show the association.

Response: Thank you. We have now deleted the p values in the results.

Discussion: Line 344-353: Those who listen to radio less than a week had lower odds compared but not those who did so once a week. I think this result is very hard to justify. You argument that listening to the radio as protective for health insurance enrolment is plausible but your result does not support. May be try merging the categories 'not at all' vs 'listens to radio (less than a week or at least once a week). If you get a result where listening to the radio comes with less likelihood of non-enrollment, your current discussion can support. Alternatively, the discussion has to reflect what you have in table 2.

Response: Our study found that women who listen to the radio less than once a week have significantly lower odds of non-enrollment in health insurance compared to those who do not listen to the radio at all, while women who listen at least once a week also showed lower odds, though this association was not statistically significant in the adjusted model. These findings suggest that listening to the radio, regardless of frequency, may increase awareness and understanding of health insurance, potentially facilitating enrollment. This aligns with findings from Amu et al. [5] in SSA, which highlight the role of media exposure in promoting health-related behaviors. Women who listen to the radio may be exposed to health messages or campaigns emphasizing the benefits of health insurance, even with limited frequency. However, the association may also reflect underlying socioeconomic factors, as women who listen to the radio may have higher education or income levels, enabling them to better understand and afford health insurance [28]. Conversely, women who do not listen to the radio may rely on alternative sources of information, such as community health workers or family networks, to support their decision-making [30,33]. These findings underscore the importance of media exposure in shaping health insurance enrollment behaviors and suggest that targeted health communication through radio could play a critical role in increasing coverage [34,35].

Line 354-362: Getting medical help for self: distance to a health facility: is important and same old issue. Multiple 'Tyranny of distance' articles have been written. It is not so much about 'quality' and 'trust' it is more about 'access issue' due to various logistical issues.

Response: In the current study, women who reported having a big problem with the distance to health facilities had higher odds of non-enrollment in health insurance coverage compared to those who did not face such a problem. This finding highlights the critical role of access barriers in shaping health insurance enrollment in Sierra Leone. Women living far from health facilities may face logistical challenges, such as long travel times, limited transportation options, and higher transportation costs, which can hinder their ability to access health services and information about health insurance enrollment [29,33,35]. Additionally, the physical distance may deter women from seeking healthcare altogether, reducing the perceived value of health insurance as they may not see it as a practical or necessary investment if accessing healthcare remains difficult [17]. These logistical and access-related challenges underscore the importance of addressing infrastructural and transportation barriers to improve health insurance coverage, particularly for women in remote areas.

Line: 363: Regional difference is very important for any country. There may be inherent issues due to systemic marginalization of certain region. I would have thought authors provide some contextual information . There is opportunity to make this paragraph really important. Think about transport access, economic and employability opportunities, education, female literacy, gender issues, ethnic variations in the regions that are prone to non-enrolment. There is very important for authors to explore.

Response: Our results indicate that women living in the Northwestern and Northern regions had lower odds of non-enrollment in health insurance coverage compared to women in the Eastern region of Sierra Leone. This regional difference likely reflects broader systemic disparities and contextual factors influencing health insurance enrollment [30]. The Eastern region has historically faced challenges such as limited transport infrastructure, which restricts access to health facilities and health insurance enrollment points, and fewer economic and employment opportunities, which may reduce the ability of women to afford health insurance [37]. Lower levels of female literacy and education in the Eastern region may also hinder women's understanding of the benefits and processes of health insurance enrollment. Additionally, gender-related issues, such as reduced autonomy in decision-making, could further limit women's ability to prioritize health insurance [30,37]. Ethnic and cultural variations may also play a role, influencing trust in formal health systems and preferences for traditional or informal providers. In contrast, women in the Northwestern and Northern regions may benefit from better transport access, higher literacy rates, and greater exposure to health-related information, enabling them to perceive the value of health insurance and access enrollment services more readily. These findings highlight the need to address systemic marginalization and regional inequities by improving infrastructure, education, and economic opportunities in underserved areas like the Eastern region to enhance health insurance coverage.

Conclusion: Public health implications have been explained in in earlier section. I would suggest authors make the conclusion really short. For example in one paragraph: highlight key points: Very low uptake, and which groups should be the focus of interventions and end with one future research.

Response: This study highlights a very low uptake of health insurance among women in Sierra Leone (96.0% non-enrollment), underscoring a significant barrier to achieving universal health coverage. Women with higher education, employment, and access to health communication via radio were less likely to be uninsured, while those facing geographic barriers, such as distance to health facilities, and those in the Eastern region were more likely to be uninsured. These findings suggest that interventions should prioritize improving education, employment opportunities, healthcare accessibility, and targeted health communication, particularly for women in underserved regions like the East. Future research should focus on exploring cultural and financial barriers to health insurance enrollment to inform more inclusive and effective policies.

Reviewer #2: Dear authors,

Thank you for studying the article entitled “Determinants of Non-enrollment in Health Insurance among Women in Sierra Leone: a cross-sectional analysis of the 2019 Sierra Leone Demographic Health Survey.” This topic is significant, but the study has some shortcomings. I would like to share a few comments that could help improve the manuscript.

Response:

Abstract: I find that the conclusion of your abstract is very long.

Response: Health insurance non-enrollment remains critically high among women in Sierra Leone. Education, employment, and access to health information via radio were associated with increased enrollment, highlighting the need to address socioeconomic barriers and leverage mass media campaigns. Geographic and regional disparities underscore the importance of improving healthcare accessibility and implementing targeted, community-based interventions to promote health insurance uptake.

Introduction:

Your introduction is well-structured and contains all the necessary information about health insurance. I congratulate you on that. However, some remarks have been made:

• Include references for the information presented. For example: line 113: include references for the information presented on the education level of women in Sierra Leone.

Response: Women, particularly those in their reproductive years, face unique barriers to enrollment, including economic constraints, limited awareness, and sociocultural factors that restrict their access to healthcare services [9].

Sierra Leone, a low-income country in West Africa with a population of approximately 7.9 million, exemplifies these challenges. Women, who constitute 51% of the population, bear a disproportionate burden of healthcare needs, particularly related to maternal and reproductive health [10]. The country continues to face a high maternal mortality ratio, recorded at 717 deaths per 100,000 live births in 2019, and a high adolescent birth rate of 102 births per 1,000 women aged 15–19 [11]

---

## [Editor Report · Decision Letter 2]

Factors associated with non-enrollment on health insurance among women in Sierra Leone: a cross-sectional analysis of the 2019 Sierra Leone demographic health survey

PONE-D-24-25104R2

Dear Dr. Osborne,

We’re pleased to inform you that your manuscript has been judged scientifically suitable for publication and will be formally accepted for publication once it meets all outstanding technical requirements.

Kind regards,

Emmanuel O Adewuyi, BPharm, MPH, PhD

Academic Editor

PLOS ONE

Additional Editor Comments (optional): Authors have appropriately addressed all comments. I am happy to recommend it for acceptance.
---

## [Editor Report · Acceptance letter]

PONE-D-24-25104R2

PLOS ONE

Dear Dr. Osborne,

I'm pleased to inform you that your manuscript has been deemed suitable for publication in PLOS ONE. Congratulations! Your manuscript is now being handed over to our production team.

Kind regards,

on behalf of

Dr. Emmanuel O Adewuyi

Academic Editor

PLOS ONE